# Tribological Evaluation of Vegetable Oil/MoS_2_ Nanotube-Based Lubrication of Laser-Textured Stainless Steel

**DOI:** 10.3390/ma16175844

**Published:** 2023-08-26

**Authors:** Marjetka Conradi, Bojan Podgornik, Maja Remškar, Damjan Klobčar, Aleksandra Kocijan

**Affiliations:** 1Institute of Metals and Technology, Lepi Pot 11, 1000 Ljubljana, Slovenia; bojan.podgornik@imt.si (B.P.); aleksandra.kocijan@imt.si (A.K.); 2Jozef Stefan Institute, Jamova 39, 1000 Ljubljana, Slovenia; maja.remskar@ijs.si; 3Faculty of Mechanical Engineering, University of Ljubljana, Aškerčeva 6, 1000 Ljubljana, Slovenia; damjan.klobcar@fs.uni-lj.si

**Keywords:** tribology, stainless steel, vegetable oil lubrication, MoS_2_ nanotubes

## Abstract

In the present work, the functionalisation of austenitic stainless steel, AISI 316L surfaces via nanosecond Nd:YAG laser texturing in order to modify the surface morphology with crosshatch and dimple patterns is presented. A tribological analysis under lubrication with sunflower and jojoba oil with and without the addition of a solid lubricant, MoS_2_ nanotubes, was performed. In conjunction with friction/wear response laser-textured surface wettability, oil spreadability and oil retention capacity were also analysed. It was shown that the crosshatch pattern generally exhibited lower friction than the dimple pattern, with the addition of MoS_2_ nanotubes not having any significant effect on the coefficient of friction under the investigated contact conditions. This was found in addition to the better oil spreadability and oil retention capacity results of the crosshatch-textured surface. Furthermore, texturing reduced the wear of the stainless-steel surfaces but led to an approximately one order of magnitude larger wear rate of the steel counter-body, primarily due to the presence of hard bulges around the textured patterns. Overall, the crosshatch pattern showed better oil retention capacity and lower friction in combination with different vegetable oils, thus making it a promising choice for improving tribological performance in various environmentally friendly applications.

## 1. Introduction

Mineral-oil-based lubrication is still the leading choice for reducing friction and wear in most industrial and commercial applications, such as in metalworking fluids [1]. However, due to the ecology-related challenges of mineral oils, such as toxicity and non-biodegradability, they represent a high risk for environmental pollution. Therefore, the research is nowadays focusing on finding suitable substitutes for mineral oils [2]. It has been shown that vegetable-oil lubrication represents a competitive substitute for mineral-oil lubrication due to its renewable, environmentally friendly and cost-effective nature [3]. In addition, its effectiveness is also reflected in its high viscosity index, good lubricity and low volatility [2,3].

Lately, several studies have focused on the introduction of specific lubricant additives to additionally reduce friction and wear during lubrication [4]. It has been shown that the efficiency of lubrication is related to the size, shape and concentration of additives [4]. Various organic and inorganic nanomaterials, such as graphene, ionic liquids, hydrogels, hexagonal BN, different metallic oxides, Cu and MoS_2_ have been considered for improving sliding efficiency and durability in cases where liquid lubricants do not meet the advanced requirements of a given application [3,5,6]. In addition, they also reduce the weight and simplify lubrication [7,8]. Nowadays, MoS_2_ lubricant films are widely used in different demanding applications, such as in aerospace, and in space industry or launch vehicles [9]. Burnishing is widely applied to MoS_2_ or other solid lubricants in order to improve the tribological properties of the roughened substrates [10,11]. MoS_2_ nanotubes have also been tested in the water-soluble, highly hydrophilic, polymer polyethylene oxide (PEO), which is used in medicine, cosmetics, as a coating for surfaces in aqueous or anhydrous media, as well as a solid polymer electrolyte in batteries and colour displays. The addition of nanotubes extends the durability of the layers by reducing wear by more than 70%. The nanotubes are effectively lubricated even after the polymer layer is already worn out. During the friction process, the nanotubes exfoliate into thin sheets that cover the surface of the steel and reduce friction for a long time [12].

A step forward in the tribological response is represented by the capability of the base material for lubricant retention [13]. The performance of the base material can be improved by surface modification via the introduction of micro/nanoscale topographies that are a result of micro/nanoscale fabrication techniques. Lee et al. [14] reported on the electrochemical etching of 304 stainless steels, resulting in porous hierarchical structures with noticeable oil retention. Hao et al. [15] studied the improvement in tribological properties after surface micro-texturing using chemical solutions. It was also shown theoretically that biomimetic texture morphologies significantly improve the tribological response through increased oil retention [16,17,18]. In the past decade, surface laser texturing has been recognised as an effective technique in enhancing the tribological performance of materials [19,20,21,22] due to its short processing times, good control over the process and friendliness to the environment. It was already shown that the texturing geometry and direction of texturing with regard to sliding during application significantly influence the lubrication efficiency and wear behaviour [23,24]. Various texturing geometries, such as micro-channels and dimples of different sizes and depths, have already been established as traps for wear debris and lubricant reservoirs, leading to reduced abrasion [22,25]. The added value of such modified surfaces is attributed to the possibility of continuous lubrication due to the lubricant retention on the textured surface, the generation of micro-hydrodynamic effects, reducing the abrasive wear due to the entrapment of wear debris and the minimisation of the contact area [26].

To combine the advantages of surface laser texturing and the use of oil/MoS_2_ nanotube-based lubrication, in this paper, an alternative approach for tribologically demanding environments with a reduced risk for environmental pollution is proposed. The friction and wear response of two different laser-textured AISI 316L morphologies, crosshatch and dimples, in vegetable oils (jojoba and sunflower) upon the addition of a solid lubricant, MoS_2_ nanotubes, were investigated and compared. In addition to the tribological analysis, the surface wettability, oil spreadability and oil retention capacity were also analysed. The obtained results offer a comprehensive insight into the friction and wear behaviour of laser-textured surfaces under oil lubrication, with or without MoS_2_ nanotube addition.

## 2. Materials and Methods

*Materials*―The steel sheet, AISI 316L, with a thickness of 1.5 mm was cut into discs of a 25 mm diameter. Prior to the laser texturing, the steel discs were hand-ground, using a grinding paper of 600 grit to remove the surface oxide layer. They were then cleaned using cotton wool and isopropanol to remove surface impurities. The average surface roughness of the prepared steel discs was Sa = (0.19 ± 0.01) µm. The sunflower and jojoba oil (supplier Tovarna Organika Ltd., Ljubljana, Slovenia) were used for lubricated tribological testing. The kinematic viscosity measured at 20 °C was 50.5 mPa·s for the sunflower oil and 33.6 mPa·s for the jojoba oil.

The MoS_2_ nanotubes were synthesised by Nanotul Ltd., Ljubljana, Slovenia following the procedure described in [12]. Their diameter was 100–200 nm and their length up to 5 µm. A transmission electron microscopy (TEM, JEOL Ltd., Tokio, Japan) image of MoS_2_ nanotubes is presented in Figure 1. Prior to tribological testing, 2 wt. % of MoS_2_ nanotubes were dispersed in both media, sunflower and jojoba oil. The mixture was alternately put into an ultrasound for 20 min and then manually shaken prior to use.

*Surface laser texturing*―A Starmark laser texturing machine was used for the production of AISI 316L test coupons. A Rofin SMD 50 W II Nd-YAG laser power source equipped with an F-Theta-Ronar lens with a focusing length of 160 mm was used for texturing. The laser’s focal point was set on the surface of the test coupons. The programming of shapes was conducted using Rofin LaserCAD software, version V7.58 where a set of crosshatch and dimple patterns were processed (Figure 2). The lines in the crosshatch texture were 15 µm deep and 50 µm wide with a spacing of 100 µm. The dimples with a diameter of 50 µm were 20 µm deep and were arranged in a pattern with a centre-to-centre distance of 100 µm. The laser texturing of dimples was carried out in pulsed mode using a pulse length of 0.06 ms, pulse frequency of 500 Hz and an electrical current of 48.0 A. Each dimple was processed with 10 pulses. The laser texturing of the lines and crosshatch pattern was conducted in CW mode with a speed of 10 mm/s and an electrical current of 42.0 A. The crosshatch pattern was conducted at a wobble frequency of 10 kHz in one run. The texturing times of the test coupons were 45 min for dimples, 14 min for the line pattern and 30 min for the crosshatch pattern. Prior to laser texturing, the surface oxide layer from the surfaces of the test samples was removed by hand grinding, using a grinding paper of 600 grit. After grinding, the surface impurities were cleaned using cotton wool and isopropanol. The laser texturing was conducted in an air atmosphere at room temperature.

*Surface characterisation*―A JEOL JSM-6500F scanning electron microscope (SEM, JEOL Ltd., Tokio, Japan) using field emission was employed to investigate the morphology of the laser-textured surfaces. Optical 3D metrology system, model Alicona Infinite Focus (Alicona Imaging GmbH, Raaba, Austria) and IF-MeasureSuite (Version 5.1) software were used to analyse the surface topography.

*Wettability, spreadability and oil-retention test*―The wettability of the laser-textured AISI 316L surface was evaluated with sunflower oil contact-angle measurements. Droplets of oil with V = 5 µL were deposited on at least three different spots on the surface to avoid the influence of roughness and gravity on the shape of the droplet. Advex Instruments s.r.o., Brno-Komín, Czechia was used to analyse the droplets and determine the contact angles. Spreadability was further analysed after 60 s by measuring the spreading area of the droplet. Oil retention capacity was determined by the application of a 500 µL oil droplet on the surface using the rotational motion of a spin coater (Laurell technologies corporation, model WS-650MZ-23NPPB). The weights of the sample before and after the application of the oil droplet were compared to further calculate the oil retention capacity via [18]
$$ORC=1-\frac{m_{1}{-m}_{2}}{m_{1}{-m}_{0}}*100\%\;$$
where *m*_0_ is the weight of the laser-textured specimen, *m*_1_ is the weight of the specimen and the oil droplet before rotation and *m*_2_ is the weight of the specimen and the oil after rotation.

*Tribological testing*―Tribological testing using a ball-on-flat contact configuration was performed under a reciprocating sliding motion on a TRIBOtechnic friction testing tribometer. One set of wear-test parameters was selected, corresponding to contact conditions experienced in critical components operating under boundary lubrication. Experiments were carried out in room conditions (RH = 50%, T = 20 °C), with a normal load of 5 N, corresponding to a nominal contact pressure of 1.2 GPa, and an average sliding speed of 5 mm/s (frequency 0.25 Hz and amplitude 10 mm). The same sliding distance of 1 m was kept for all the tests for the purpose of a relative-wear comparison between the different texturing parameters and the orientation. Each test was repeated at least three times in order to obtain statistically relevant results. A 100Cr6 bearing steel ball with a diameter of 5 mm (Ra = 0.05 μm, 58 HRC) was used as a stationary counter-body and loaded against the as-textured AISI 316L disc without any post-polishing or additional processing. A relatively small ball diameter of 5 mm was selected to simulate the most critical contact situations with very small contact areas (edges) and high-pressure spikes and to examine whether the texturing type, spacing and orientation still influence the tribological behaviour or whether it is overwhelmed by the small contact size. The CoF was measured continuously during the tests, with the steady-state coefficient of friction being determined as the average value of the last 100 s. The wear volume of the textured surfaces could not be measured; therefore, the wear tracks of the counter body were analysed. 

## 3. Results

### 3.1. Surface Morphology

The AISI 316L samples were hand-ground with 600-grit grinding paper and textured using a LPKF nanosecond Nd-YAG marking laser to modify the surface with crosshatch and dimple patterns. Figure 3 shows the SEM images of the details of the laser texturing patterns and the corresponding height profiles that define the morphological details, i.e., the crosshatch texture and the dimples. The width of the channels in the crosshatch-textured samples is approximately 50 µm and the depth is approximately 10 µm. It can also be observed that the laser produces a certain amount of ejected material on both sides of the laser channel, with the height of the bulges extending up to 15 μm. The diameter of the dimples is approximately 50 μm, while the maximum depth of the dimples is around 20 µm. The bulges of the ejected material are also observed at the edge of the circle, however, not as high as in the case of the crosshatching-textured samples, extending up to 5 μm. Furthermore, the laser texturing not only affects the surface morphology, but also the microstructural-crystallographic characteristics. Significant surface hardening (~310–315 HV0.01) was observed for the melted and re-solidified surface of both laser textured samples in comparison to the base material with a surface hardness ~240 HV0.01.

### 3.2. Tribological Analysis

#### 3.2.1. Coefficient of Friction

Steady-state coefficient of friction (COF) values are shown in Figure 4. In the case of dry-sliding in air, the texturing results in an increased steady-state COF, increasing it from 0.4 to about 0.6, with the crosshatch pattern showing slightly higher values. An increase in friction can be attributed to the reduced contact area and abrasive action of the hard bulges formed around the textured patterns, being more pronounced for the crosshatch pattern. However, in the case of lubricated contact, laser texturing provides more than 50% lower friction with respect to the surfaces without laser texturing. Texturing provides micro-hydrodynamic effects, with the lubrication regime changing from boundary/mixed into hydrodynamic, as indicated by the level of the coefficient of friction [27]. For reference, un-textured surface steady-state COF is between 0.25 and 0.3 and for the textured ones it is around 0.1. Regardless of the oil used, the crosshatch pattern shows a 10–20% lower friction, while the addition of MoS_2_ nanotubes has no effect under the contact conditions investigated. For all oils used, a similar friction trend was observed, with the initial coefficient of friction being in the range of 0.15 and then reaching the steady-state conditions in 100 s (Figure 5). This indicates that the whole friction-reducing effect comes from the texturing and not from the oil or additive used. 

Further on, the static contact angles of the sunflower oil and sunflower oil+MoS_2_ on un-textured and laser-textured samples were also evaluated. For sunflower oil, it was observed that for both textures, crosshatch and dimples, the contact angles were below 8° ± 2°, while for the un-textured sample, the angle was 10° ± 2°. The spreadability test was performed after 60 s. The circular-shaped spreading area of the droplet was observed on the un-textured and dimple-textured surface, with sizes of around 0.4 cm^2^ and 1.3 cm^2^, respectively. On the other hand, an elliptical-shaped spreading area of the droplet was observed on the crosshatch-textured surface, sized 1.7 cm^2^ (Figure 6a). The oil retention capacity in correlation with COF is shown in Figure 6b. The highest ORC, 1.10%, was observed for the crosshatch-textured surface compared to 0.86% for the dimple-textured surface and to the un-textured surface with the lowest ORC of 0.21%. The significantly better oil spreadability and oil retention capacity of textured surfaces also result in a considerably lower coefficient of friction (0.1 vs. 0.3). Furthermore, better oil spreadability and oil retention capacity also indicate the better tribological performance of the crosshatch-textured compared to the dimple-textured surface, resulting in a 10–20% lower steady-state COF for the crosshatch-textured surface. A similar trend was observed for sunflower oil+MoS_2_, showing a similar contact angle and spreadability, but a 30% reduced ORC for textured surfaces (0.78% for crosshatch and 0.58% for dimples), which is compensated by the increased viscosity, thus resulting in similar COF values as for pure oil. The ORC for the un-textured sample under sunflower oil+MoS_2_ lubrication was about the same as for pure sunflower oil, 0.22%, while the spreadability was reduced to the circular-shaped area sized around 0.2 cm^2^. Similar values and trends were observed also for jojoba oil and jojoba oil+MoS_2_. However, jojoba oil with lower viscosity shows a lower ORC but slightly better oil spreadability, thus resulting in about 10% lower steady-state friction.

#### 3.2.2. Wear

The wear volume of the textured stainless-steel disc samples could not be determined; therefore, the wear volume of the 100Cr6 steel counter-ball was analysed, being determined as the volume of the removed spherical section. As shown in Figure 7 and Figure 8, texturing reduces the wear of the stainless-steel disc, but due to hard bulges formed around the textured pattern it results in about a one order of magnitude larger wear of the counter-body (Table 1 and Table 2). In all cases, the main wear mechanism is sliding abrasive wear, combined with the minor adhesive wear component. However, when switching from non-textured to textured surfaces, the textured pattern with bulges intensifies the abrasive wear component. In general, for all contact surfaces (un-textured and textured), the use of jojoba oil in general results in lower counter-body wear, which can be related to the better oil spreadability of the jojoba oil as compared to sunflower oil. However, the addition of MoS_2_ nanotubes has the opposite effect, reducing the oil retention capacity (30% lower) of both oils, thus resulting in increased counter-body wear. This effect is even more pronounced for textured contact, with nanotubes reducing the micro-hydrodynamic effects of the textured pattern. A comparison of the crosshatch- and dimple-textured patterns reveals increased counter-body abrasive wear for the crosshatch pattern. Although the crosshatch pattern shows better oil retention capacity, it is characterised by denser and more pronounced bulges, thus causing intensified counter-body abrasive wear. 

## 4. Conclusions

The surface morphology of the AISI 316L samples was modified through laser surface texturing, resulting in the creation of crosshatch and dimple patterns with characteristic bulges of the surface-hardened ejected material around the textures. 

Within the tribological analysis, the lubricated sliding behaviour in two vegetable oils, jojoba and sunflower, with or without the addition of MoS_2_ nanotubes was compared. Under lubricated conditions, the textured surfaces demonstrated more than 50% lower friction as compared to un-textured ones, indicating micro-hydrodynamic effects and a shift towards a hydrodynamic lubrication regime. The un-textured surface steady-state COF was between 0.25 and 0.3 and around 0.1 for the textured surfaces. Regardless of the oil used, the crosshatch pattern showed a 10–20% lower friction than the dimple pattern, with the addition of MoS_2_ nanotubes not having any significant effect on the COF under the investigated contact conditions. A possible reason is the relatively short length of the MoS_2_ nanotubes in comparison with the deepness of the laser-texturing channels, which limits their contribution to the friction contact. The improved friction behaviour of the crosshatch pattern was supported by its better oil spreadability and oil retention capacity results. Furthermore, lubrication under jojoba oil with a lower viscosity, lower ORC but better spreadability also resulted in a lower COF than for sunflower oil. 

Texturing reduced the wear of the stainless-steel disc but led to an approximately one order of magnitude larger wear rate of the steel counter-body, primarily due to the presence of hard bulges around the textured patterns. The main wear mechanism was sliding abrasive wear, combined with a minor adhesive wear component. When switching from non-textured to textured surfaces, a textured pattern with bulges intensifies the abrasive wear component. The use of jojoba oil with a better spreadability effect generally resulted in lower counter-body wear compared to sunflower oil. However, the addition of MoS_2_ nanotubes reduced ORC and increased counter-body wear, mainly by reducing the micro-hydrodynamic effects of the textured surface. The highest ORC, 1.10%, was observed for the crosshatch-textured surface compared to 0.86% for the dimple-textured surface and compared to the un-textured surface with the lowest ORC of 0.21%.

In summary, the crosshatch pattern exhibits better oil retention capacity and lower friction in combination with different vegetable oils, thus making it a promising choice for improving tribological performance in various environmentally friendly applications.

## Figures and Tables

**Figure 1 materials-16-05844-f001:**
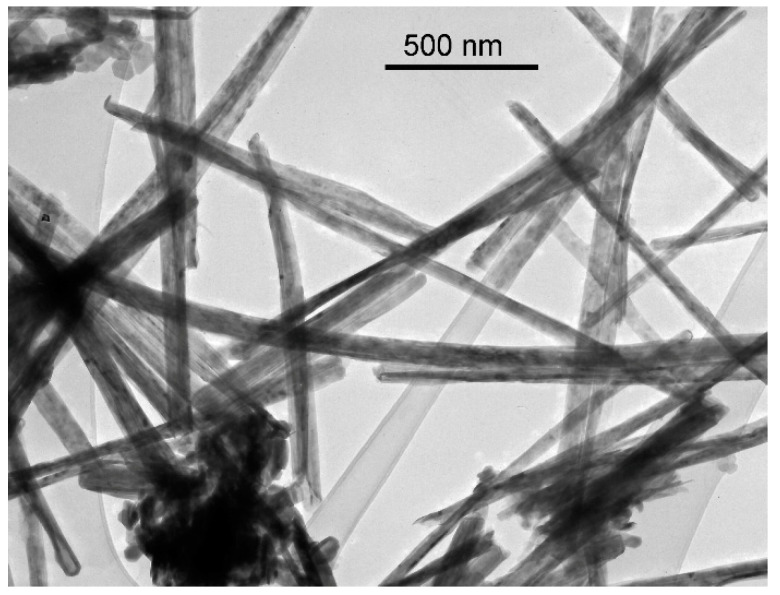
TEM image of MoS_2_ nanotubes.

**Figure 2 materials-16-05844-f002:**
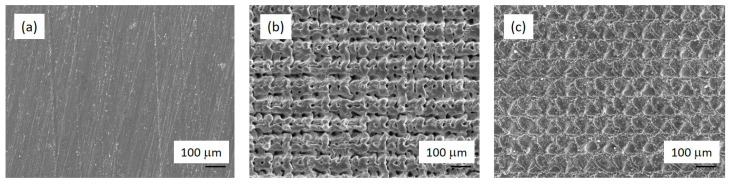
SEM micrographs of un-textured surface (**a**) and laser-textured surfaces, crosshatch (**b**) and dimples (**c**) patterns.

**Figure 3 materials-16-05844-f003:**
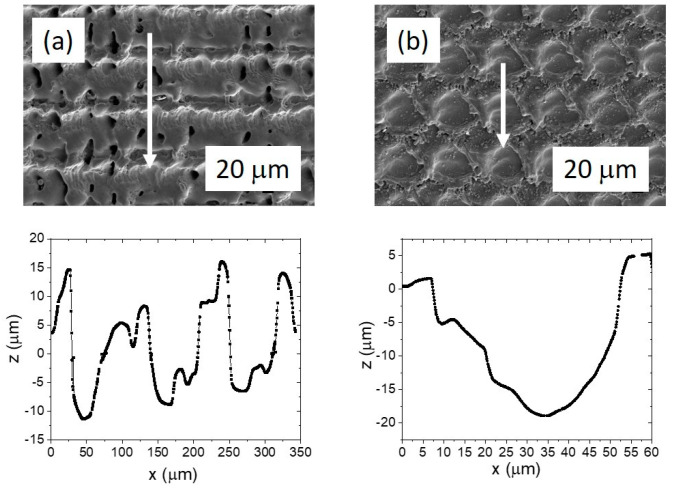
SEM images of the details of laser-texturing channels in the crosshatching-textured samples (**a**) and the dimple-textured surface (**b**). The arrows on the SEM images indicate the direction of the profile measurement, which is shown below each image. The text continues here (Figure 2 and Section 3.2.2).

**Figure 4 materials-16-05844-f004:**
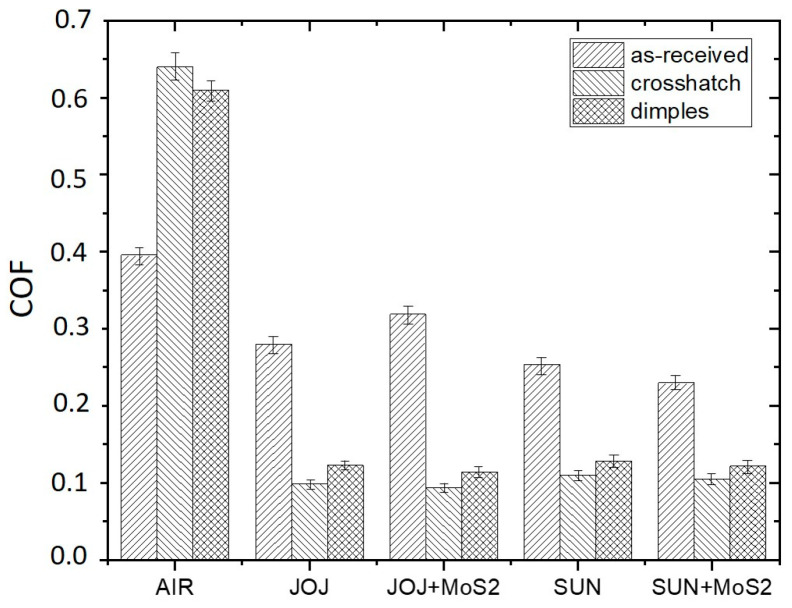
Steady-state coefficient of friction for as-received and laser-textured samples in air and in jojoba/sunflower oil without and with the addition of MoS_2_ nanotubes.

**Figure 5 materials-16-05844-f005:**
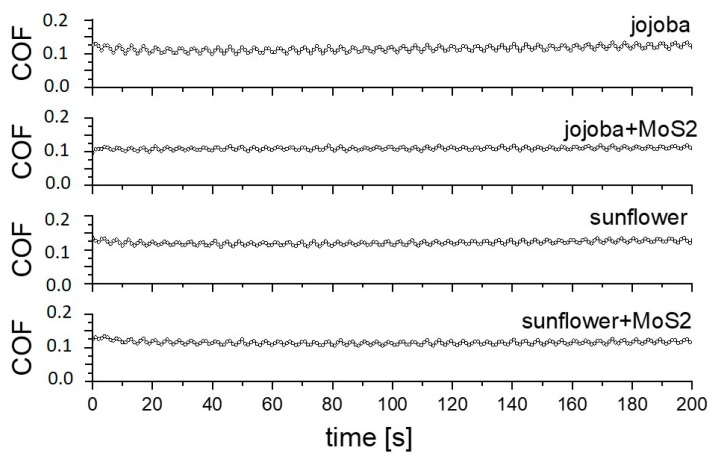
Coefficient of friction curves for lubricated dimple-textured surface.

**Figure 6 materials-16-05844-f006:**
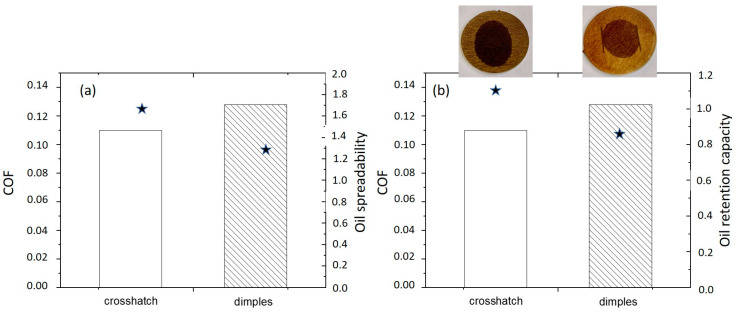
The relationship between the COF and oil droplet spreadability (star) (**a**) and the COF and oil retention capacity (star) (**b**) in the sunflower oil. The inset images in (**b**) show the shape of the oil droplets after 60 s, circular vs. elliptical.

**Figure 7 materials-16-05844-f007:**
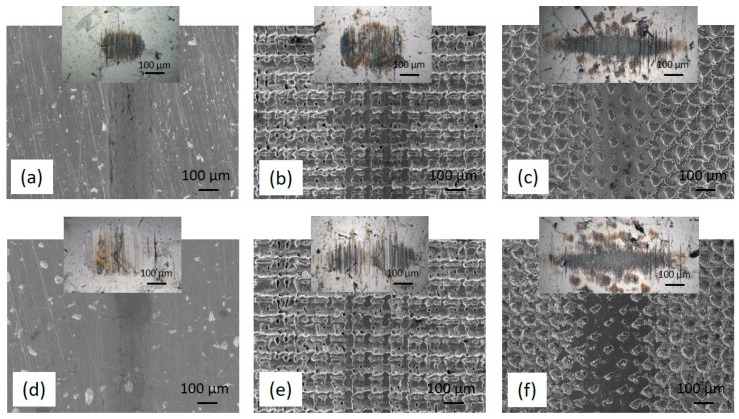
SEM images of the wear tracks obtained in jojoba oil without (**a**–**c**)/with (**d**–**f**) MoS_2_ nanotubes: as-received (**a**,**d**) sample, crosshatch- (**b**,**e**) and dimple-textured (**c**,**f**) samples. The insets in the images show the worn surface of the sliding ball.

**Figure 8 materials-16-05844-f008:**
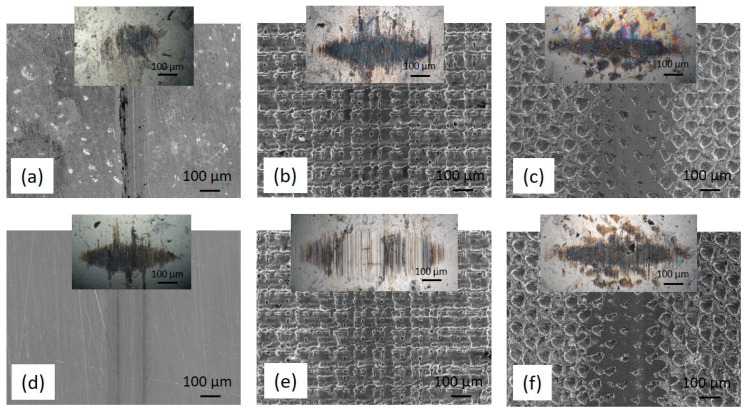
SEM images of the wear tracks obtained in sunflower oil without (**a**–**c**)/with (**d**–**f**) MoS_2_ nanotubes: as-received (**a**,**d**) sample, crosshatch- (**b**,**e**) and dimple-textured (**c**,**f**) samples. The insets in the images show the worn surface of the sliding ball.

**Table 1 materials-16-05844-t001:** Counter-body wear track dimensions—jojoba oil.

	Long Axis [µm]	Short Axis [µm]	Wear Volume [mm^3^]
as-received jojoba	227 ± 10	164 ± 10	(0.28 ± 0.02) × 10^−4^
as-received jojoba+MoS_2_	247 ± 11	212 ± 19	(0.54 ± 0.03) × 10^−4^
crosshatch jojoba	336 ± 18	250 ± 16	(1.44 ± 0.08) × 10^−4^
crosshatch jojoba+MoS_2_	461 ± 19	205 ± 15	(2.27 ± 0.09) × 10^−4^
dimples jojoba	657 ± 22	151 ± 11	(5.22 ± 0.51) × 10^−4^
dimples jojoba+MoS_2_	699 ± 23	145 ± 10	(6.25 ± 0.55) × 10^−4^

**Table 2 materials-16-05844-t002:** Counter-body wear track dimensions—sunflower oil.

	Long Axis [µm]	Short Axis [µm]	Wear Volume [mm^3^]
as-received sunflower	240 ± 18	178 ± 12	(0.37 ± 0.03) × 10^−4^
as-received sunflower+MoS_2_	577 ± 20	176 ± 15	(3.95 ± 0.29) × 10^−4^
crosshatch sunflower	651 ± 25	199 ± 18	(6.39 ± 0.45) × 10^−4^
crosshatch sunflower+MoS_2_	719 ± 30	241 ± 23	(10.45 ± 0.91) × 10^−4^
dimples sunflower	655 ± 21	151 ± 12	(5.17 ± 0.39) × 10^−4^
dimples sunflower+MoS_2_	695 ± 25	149 ± 12	(6.22 ± 0.55) × 10^−4^

## Data Availability

There is still further research going on and the data are still in use.

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
