# Peer review of "Tribological Evaluation of Vegetable Oil/MoS2 Nanotube-Based Lubrication of Laser-Textured Stainless Steel"

_materials, 2023, doi:10.3390/ma16175844_

Round 1

Reviewer 1 Report

  • The introduction is written very badly. The title of the work contains “vegetable oil/MoS2 nanotubes” as key words. Thus, it is the role of the MoS2 nanotubes that should be considered in detail in the introduction with particular attention to length of nanotubes and influence on ORC and hydrodynamics. And what about other nanotubes, e.g., oxides of metals?

  • The aim of the work should be described more clearly. The actual novelty should be emphasized more distinctly. What does the additive of MoS2? How exactly does the MoS2 additive improve the tribological properties on example of other similar systems described in literature? Lines 36-39 are absolutely insufficient.

  • There are many typos, incorrect designations and spelling variants.

  • Fig. 6 caption is unclear and very bad.

 English is quite clear

Author Response

  • The introduction is written very badly. The title of the work contains “vegetable oil/MoS2 nanotubes” as key words. Thus, it is the role of the MoS2 nanotubes that should be considered in detail in the introduction with particular attention to length of nanotubes and influence on ORC and hydrodynamics. And what about other nanotubes, e.g., oxides of metals?

Thank you for your comment. Your suggestion was implemented in the introduction.

  • The aim of the work should be described more clearly. The actual novelty should be emphasized more distinctly. What does the additive of MoS2? How exactly does the MoS2 additive improve the tribological properties on example of other similar systems described in literature? Lines 36-39 are absolutely insufficient.

Thank you for your comment. Your suggestion was implemented in the introduction.

  • There are many typos, incorrect designations and spelling variants.

Thank you for your valuable comment. The text was checked again.

  • 6 caption is unclear and very bad.

Thank you for your valuable comment. The text was corrected.

Reviewer 2 Report

1. Many places it was used as “we presented”, “We performed” etc. These need to be corrected grammatically

2. Don’t use a paragraph for two or three lines. There should be a minimum of 6 lines in a paragraph. Please modify accordingly in section 2.

3. Suggested to include photographs of samples considered (before and after texturing) in Section2.

4. Check line number 73 : 20 oC

5. Suggested to justify why MoS2 nanotubes were considered for the present study

6. Suggested to incorporate the photograph of laser texturing setup (while performing texturing)

7. Suggested to include tribology setup (photograph) considered in the present study

8. Wear testing parameters and their variation need to be discussed in detail

9. Friction variation as a function of time need to be presented

10. Suggested to evaluate specific wear rate rather than wear volume.

11. More details about wear mechanism need to be discussed along with SEM images of wear tracks

12. Applications for which the current research performed need to be highlighted clearly

13. Suggested to perform tensile test for the considered specimens for basic mechanical properties of the considered specimen.

Many errors were observed throughout the manuscript. 

Author Response

Reviever 2

  1. Many places it was used as “we presented”, “We performed” etc. These need to be corrected grammatically

Thank you for your valuable comment. The text was corrected.

  1. Don’t use a paragraph for two or three lines. There should be a minimum of 6 lines in a paragraph. Please modify accordingly in section 2.

Thank you for your valuable comment. It was corrected.

  1. Suggested to include photographs of samples considered (before and after texturing) in Section2.

Thank you for your valuable comment. The image of the un-textured surface was added in Fig. 2a.

  1. Check line number 73 : 20 oC

Thank you for your valuable comment. The text was corrected.

  1. Suggested to justify why MoS2nanotubes were considered for the present study

Thank you for the comment. MoS2 nanotubes are known for its excellent lubrication properties and our intention was to combine oil/solid lubrication to further improve tribological characteristics. The text was improved accordingly.

  1. Suggested to incorporate the photograph of laser texturing setup (while performing texturing)

Thank you for your valuable comment. This is a standard setup and including additional image would increase the number of Figures above the allowable number.

  1. Suggested to include tribology setup (photograph) considered in the present study

Thank you for your valuable comment. This is a standard setup and it was already presented in our previous work, i.e. https://doi.org/10.1680/jsuin.21.00048

  1. Wear testing parameters and their variation need to be discussed in detail

Thank you for the comment. One set of wear test parameters was selected, corresponding to contact conditions experienced in critical components operating under boundary lubrication. This description is added into the manuscript.

  1. Friction variation as a function of time need to be presented

Thank you for your valuable comment. Friction variation as a function of time is presented in Fig. 5 and described in chapter 3.2.1

“For all oils used similar friction trend was observed, with the initial coefficient of friction being in the range of 0.15 and then reaching the steady-state conditions in 100 s (Fig. 5).”

  1. Suggested to evaluate specific wear rate rather than wear volume.

Thank you for the comment. In this specific case focus is on the comparison between different oils and addition of MoS2, all tested under the same contact conditions. Evaluating wear rate would be important, when comparing tests under different contact conditions (load, sliding distance). However, in this particular case trends are completely the same either evaluating wear volume or wear rate. Therefore, we decided to keep the wear volume results.

  1. More details about wear mechanism need to be discussed along with SEM images of wear tracks

Thank you for the comment. Main wear mechanism observed was abrasive wear with minor adhesive wear component. However, when switching from non-textured to textured surfaces bulges formed around dimples intensifies abrasive wear of the counter-body. This is now explained in the manuscript:

“In all cases main wear mechanism is sliding abrasive wear, combined with minor adhesive wear component. However, when switching form non-textured to textured surfaces, textured pattern with bulges intensifies abrasive wear component. …

Comparison of crosshatch- and dimple-textured pattern reveals increased counter-body abrasive wear for crosshatch pattern. Although crosshatch pattern shows better oil retention capacity it is characterized by denser and more pronounced bulges, thus causing intensified counter-body abrasive wear.”

  1. Applications for which the current research performed need to be highlighted clearly

Thank you for your comment. The possible applications were added to the introduction as suggested.

  1. Suggested to perform tensile test for the considered specimens for basic mechanical properties of the considered specimen.

Thank you for your valuable comment. Here, we were interested in tribological response of specific surfaces rather than in bulk properties of the specimens. Therefore, tensile tests were not considered.

Round 2

Reviewer 2 Report

(i) Anova need to be performed for the obtained results

(ii) More results need to be incorporated with statistical analysis

(iii) Conclusions need to be more specific 

NIL

Author Response

  • Anova need to be performed for the obtained results

Thank you for your comment. Statistical evaluation was added to the results where applicable.

  • More results need to be incorporated with statistical analysis

Thank you for your valuable comment. All friciton experiments wer performed at least three times. This was added to the manuscript.

  • Conclusions need to be more specific 

Thank you for your valuable comment. We improved the conclusions as suggested.